# Endpoints-Clipping CSI Amplitude for SVM-Based Indoor Localization

**DOI:** 10.3390/s19173689

**Published:** 2019-08-25

**Authors:** Zhanjun Hao, Yan Yan, Xiaochao Dang, Chenguang Shao

**Affiliations:** 1College of Computer Science and Engineering, Northwest Normal University, Lanzhou 730070, China; 2Gansu Province Internet of Things Engineering Research Center, Lanzhou 730070, China

**Keywords:** Channel State Information, indoor localization, fingerprint positioning, DBSCAN, SVM

## Abstract

With the wide application of Channel State Information (CSI) in the field of sensing, the accuracy of positioning accuracy of indoor fingerprint positioning is increasingly necessary. The flexibility of the CSI signals may lead to an increase in fingerprint noise and inaccurate data classification. This paper presents an indoor localization algorithm based on Density-Based Spatial Clustering of Applications with Noise (DBSCAN), Endpoints-Clipping (EC) CSI amplitude, and Support Vector Machine (EC-SVM). In the offline phase, the CSI amplitude information collected through the three channels is combined and clipped using the EC, and then a fingerprint database is obtained. In the online phase, the SVM is used to train the data in the fingerprint database, and the corresponding relationship is found with the CSI data collected in real time to perform matching and positioning. The experimental results show that the positioning accuracy of the EC-SVM algorithm is superior to the state-of-art indoor CSI-based localization technique.

## 1. Introduction

With the development of the Internet of Things and popularity of intelligent terminal devices, wireless indoor positioning technology has been researched in different ways [1,2]. At present, indoor positioning technologies appearing at home and abroad generally include Global Positioning System (GPS), Radio Frequency Identification (RFID), and fingerprinting positioning technology. GPS has the advantages of rapid positioning, accuracy and stability in a spacious outdoor environment [3,4]. However, due to factors such as occlusion in indoor conditions, the positioning accuracy is seriously degraded or even impossible to locate. RFID technology uses radio frequency for noncontact data exchange to achieve identification and positioning, it has the advantages of large transmission range and low equipment cost [5,6]. However, because this technology does not have communication capability, it is not conducive to integration into other systems. Therefore, passive fingerprint positioning technology has been proposed, which does not require carrying extra devices to locate the target in positioning phase. However, due to the instability and inactivity of Received Signal Strength (RSS) signal, the positioning accuracy is affected. In terms of access, the RSS signal is taken from the MAC layer, there is no uniform standard for the RSS signals of various manufacturers, the same power may have different values on the Wi-Fi devices of different manufacturers [7], and the Channel State Information (CSI) is taken from the physical layer [8]. Theoretically, CSI signal can provide more detailed channel information, so it captures more environmental information for positioning [9]. Therefore, the indoor positioning technology based on CSI has attracted increasing attention in recent years.

The traditional Wi-Fi fingerprint positioning technology usually uses the RSS signal before the CSI can be extracted from Wi-Fi NICs. First, the form of the positioning is the collected data to construct a fingerprint database as the feature value of the location, and then the sampled data is compared with the fingerprint database to obtain the positioning result, such as RADAR [10], Ekahau, Horus [11], etc.; because of the limitations of the RSS signal, the positioning accuracy ranges from 2 to 5 m. RSS signal can be obtained from ordinary Wi-Fi receiving devices, which directly reflects the power strength from the transceiver, and maintains the influence of Wi-Fi signal in the indoor positioning field to a certain extent [12]. However, for CSI, the RSS only indicates the total received energy of the channel, and does not characterize the environmental characteristics such as multipath in detail. Therefore, the CSI fingerprint matching method has more potential compared with the characteristics of RSS. After acquiring the CSI signal, researchers use CSI to complete fingerprint database establishment, matching and positioning for the first time. Indoor positioning systems based on CSI signals typically use information from multiple links for positioning. Zhu et al. [13] proposed a CSI indoor positioning method based on Bayesian filtering method. Bayesian filtering is used for data processing to reduce the time variation of the received signal. Wu, et al. [14] used the naive Bayesian algorithm for position classification, and also reduced position misjudgment by combining the results of multiple antenna pairs. A passive indoor positioning technology based on CSI and naive Bayes is proposed in the work by the authors of [15], which improves the naive Bayesian algorithm to make the positioning result more accurate.The experimental part verifies the performance of the positioning algorithm in a variety of environments. Wu et al. [16] proposed a passive indoor positioning algorithm combined with mathematical statistics and networks, mainly by improving the visibility graph (VG) technology to establish a complex indoor network to achieve better positioning accuracy. Xiao et al. [17] implemented the FIFS system and used CSI data for fingerprint database establishment; the system utilizes multiple CSI measurements at one time and is more stable over time, causing the positioning accuracy to be better than Horus [11]. However, FIFS only uses the amplitude feature, but simply performs the simple averaging method on the obtained amplitude value. Gjengset et al. [18] implemented the phaser system based on commercial network cards and proposed a method for automatically calibrating phase errors. Kotaru et al. [19] and others implemented SpotFi, using existing common network cards, and finally achieved better positioning results. However, its positioning stage only uses phase information in the works by the authors of [18,19]. Zhou et al. [20] built a fingerprinting database based on the features of amplitude and phase; however, the K-Nearest Neighbor (KNN) positioning method is finally used for positioning, resulting in low positioning accuracy. Wang, et al. [21] realized the deep learning technology introduced into the DeepFi system and improved the positioning accuracy. The deep learning technique is also applied in the work by the authors of [22], but it has made related improvements to the non-line-of-sight situation. They used neural networks for training and achieved good positioning results in works by the authors of [20,21,22], but it is training data for specific environments. Chapre et al. [23] implemented CSI-MIMO systems, taking into account the spatial diversity and frequency diversity of CSI measurements. Since signal propagation of different subcarrier frequencies is affected by the environment, CSI-MIMO subtracts the measured values of adjacent subcarriers to obtain a fingerprint. Wu et al. [24] simply removed the outliers and used the median values of different subcarriers as reference values. It is only requires a simple mathematical method to denoise the CSI data in works by the authors of [23,24], resulting in unsatisfactory positioning results. In summary, ensuring the robustness while improving the positioning accuracy is an urgent problem to be solved.

A CSI-based EC-SVM localization algorithm is proposed in this paper. In order to reduce the noise of the amplitude data, the algorithm combines the three links to find the most sensitive point to cut and form a robust fingerprint database. The SVM algorithm is used for classification to finally complete the positioning.

In general, in this work, we have made the following main contributions.
We propose a novel indoor fingerprint localization algorithm. According to the characteristics of the communication link, the collected CSI data is denoised by the Density-Based Spatial Clustering of Applications with Noise (DBSCAN) algorithm in the online and offline phases.We propose an EC noise reduction method, which firstly integrates three CSI communication links, and then performs feature extraction on the combined link to obtain a robust fingerprint database.We validate the proposed theory and method in a real experimental scenario. The experimental results show that the proposed positioning algorithm is superior to its comparison algorithm and has high robustness.

The rest of this paper is organized as follows. In Section 2, we introduce the proposed system model and relevant definitions. In Section 3, we introduce the EC-SVM indoor localization system. In Section 4, the performance of the proposed positioning method is verified by experiments. Finally, conclusions are drawn in Section 5.

## 2. System Model and Relevant Definitions

The traditional WLAN-based passive indoor positioning method is mainly based on RSS. Due to the shortcomings of the RSS signal, such as poor signal stability and poor reliability, the actual positioning effect is not ideal. Compared to RSS, CSI provides more granular channel information, which can solve the problems encountered above. CSI can be easily obtained by modifying the Atheros-CSI-tool driver code to support NIC that support the IEEE 802.11n standard at the kernel level and rewriting applications [25,26]. CSI is the channel attribute of a communication link. It describes the weakening factor of the signal in each transmission path, that is, the value of each element in the channel gain matrix H [27], such as scattering, fading, multipath fading or shadowing fading, power decay of distance, and other information. CSI can adapt the communication system to current channel conditions, providing high reliability and high rate communication in multiantenna systems.

In a smoothly attenuated channel, channel information can be modeled in the frequency domain by Orthogonal Frequency Division Multiplexing (OFDM) as
(1)$$\vec{Y}\;=H\vec{X}\;+\vec{N}\;$$
where $\vec{Y}\;$ and $\vec{X}\;$ represent the received and transmitted signal vectors, respectively. Vector $\vec{N}\;$ is Gaussian noise and H denotes the channel’s frequency respond. The CSI of all subcarriers can be expressed by the formula as
(2)$$H=\frac{\vec{Y}\;}{\vec{X}\;}$$

Current WLAN technologies, such as 802.11n, employ OFDM and Multiple-Input Multiple-Output (MIMO) technologies [28,29]; CSI is divided into different subcarrier groups. Under the bandwidth of 40 MHz, the number of subcarriers is 114, then the CSI matrix H can be expressed as
(3)$$H=\left[\begin{array}{cccc} H_{11} & H_{12} & \cdots & H_{1q}\\ H_{21} & H_{22} & \cdots & H_{2q}\\ \vdots & \vdots & \ddots & \vdots \\ H_{p1} & H_{p2} & \cdots & H_{pq} \end{array}\right]$$
where *p* and *q* represent the number of transmitting antennas and the number of receiving antennas, respectively, and m=p×q is the number of pairs of antennas. Hence, a complex value $H_{i}$ of $i_{th}$ subcarrier can be defined as
(4)$$H_{i}=\left|H_{i}\right|e^{j\sin\left(∠H_{i}\right)}$$
where $\left|H_{i}\right|$ and $∠H_{i}$ are the amplitude response and the phase response of subcarrier *i*, respectively.

According to the wireless signal propagation characteristics, the energy of the wireless signal is attenuated as the distance increases. The emergence of Wi-Fi was originally used for close high-speed communication, but with the application of OFDM technology in Wi-Fi [30]; its CSI contains multiple subcarrier data, which can provide more information to reflect the target location. Because of the occlusion of the human body to the signal, the response of the CSI signal will change, so the target can be located by this characteristic. As shown in the Figure 1, the amplitude values at different locations are different, so we can use the amplitude to locate.

Figure 1 shows 100 measurements of 114 subcarrier in channel 1 of the 3 * 1 antenna beams (including one transmitting antenna and three receiving antennas) when the communication system is in the 40 MHz bandwidth. The raw amplitude of location 1 and 2 are represented in Figure 1a,b. We can draw a conclusion from these two pictures that CSI amplitude shows different curves in different location, which also verifies the localization feasibility that CSI amplitude can be used as signal feature fingerprint.

As shown in Figure 1, the CSI amplitude values collected at different locations. It can be seen from Figure 1a,b that most of the CSI amplitude values are in a cluster of closely adjacent states, but some of the data is obviously far away from the data cluster set, and the part of the data is called abnormal value. To ensure accuracy, the DBSCAN algorithm can be used to eliminate the abnormal parts. DBSCAN is a density-based clustering algorithm. This kind of clustering algorithm generally assumes that categories can be determined by the tightness of the sample distribution. The same category of samples is closely related to each other, that is to say, there are samples of the same category not far from any sample of this category. CSI signal has a fine-grained ability of sense and it easily interfered due to multipath effect with serious signal error at the receiver, from the data object where the amplitude value is located, we can easily calculate the parameters in the DBSCAN algorithm. It can be seen from Figure 2 that the CSI amplitude data distributed in the upper part shows a high degree of correlation, indicating that the data of this part belong to one category. The CSI amplitude data in the lower part is loosely dispersed, and its full correlation cannot be guaranteed. If the part of the data cannot be attributed to the same category, it is called an outlier. Therefore, the outliers can be eliminated by the judgment of the method to ensure the accuracy of the data.

SVM was initially defined by Vladimir Vapnik in the early 1990s to resolve discrimination issues: classification and regression analysis [31,32]. The classification of SVM is mainly for the two classification of sample data. This feature is used in the design of the positioning algorithm proposed in this paper to classify the processed CSI sample data.

## 3. EC-SVM Indoor Localization System

### 3.1. System Model

The algorithm proposed includes CSI amplitude sample data and noise reduction between CSI subcarriers. As shown in Figure 3, the system model is divided into the online phase and the offline phase. The DBSCAN processing of the feature data is performed in the offline phase and the online phase. In the offline phase, the CSI amplitude sample data is collected, then the fingerprint database is established after the abnormal value is processed by the DBSCAN algorithm. In the online stage, the real-time sample data of the acquired CSI amplitude is processed by the DBSCAN algorithm to process the outliers, and then feature extraction of CSI amplitude sample data by EC method, and the SVM is used to classify according to the characteristics of the clipped CSI amplitude, and finally the physical position estimation result is obtained.

### 3.2. Noise Reduction of CSI Sample

#### 3.2.1. The Combination of the Communicating Link

The CSI amplitude sample data collected at a certain location to be located has an abnormal value due to the influence of the multipath effect in the environment. The results of the CSI data of the chain three channels using the DBSCAN algorithm to eliminate the outliers are shown in Figure 4a–c. In order to perform noise reduction processing on the CSI data of one location, we combine the amplitude values of the three links, and the result of flattening is shown in Figure 4d. Since each link contains 114 subcarriers, the flattened pattern contains 342 subcarriers.

#### 3.2.2. Charateric of Combined Communication Extract

We have a simple combination of the three data links, but although the data link we use is the DBSCAN algorithm for noise reduction, there is still noise in the data. In order to further optimize the CSI data, we propose a novel EC noise reduction method. In order to express the steps of the EC algorithm more clearly, we randomly take a CSI amplitude data from the combined CSI data link as shown in Figure 5a. We use the splicing points *a*, *b* of the link as the base point for finding the clipping point. We define the clipping factor sequences for the two points *a* and *b* as α and β, respectively:(5)$$\begin{array}{c} \alpha =\left\{\begin{array}{c} \left[\left(a-1\right),\;\left(a-2\right),\left(a-3\right),\cdots \left(a-n_{1}\right)\right],\;n_{1}\in \left(0,114\right)\\ \left[\left(a+1\right),\;\left(a+2\right),\left(a+3\right),\cdots \left(a+n_{2}\right)\right],\;n_{2}\in \left(114,228\right) \end{array}\right. \end{array}$$
(6)$$\begin{array}{c} \beta =\left\{\begin{array}{c} \left[\left(b-1\right),\;\left(b-2\right),\left(b-3\right),\cdots \left(b-n_{3}\right)\right],\;n_{3}\in \left(114,228\right)\\ \left[\left(b+1\right),\;\left(b+2\right),\left(b+3\right),\cdots \left(b+n_{4}\right)\right],\;n_{4}\in \left(228,342\right) \end{array}\right. \end{array}$$
where a−1 represents the first clipping point found to the left of point a, a+1 represents the first clipping point found to the right of point a, b−1 represents the first shear point found to the left of point b, and b+1 represents the first shear point found to the right of point b.

First, we search the clipping points on both sides of point a. Starting from point a to the left, the monotonicity of the curve is judged by the difference of adjacent points until its monotonicity changes. We take the adjacent point where monotonicity changes as the first clipping point to be found. Since the curve trend of the CSI amplitude data is wavy, we will find many eligible cilpping points. For the same reason, many cut points are found on the left and right sides of point a. However, through experimental verification analysis, we obtained that the sample data formed by the first clipping point among the many clipping points found on both sides of point a is better. After determining the clipping points on both sides of point a, we take the data between the two clipping points as the first feature sample at a certain position. In the same way, we can find the shear points on both sides of point b to obtain another feature sample at the same position. The samples obtained at two points a and b are sample features at one location. The feature sample data of the 25 positions obtained by the EC method is as shown in Figure 6. The pseudocode of the specific clipping process is given in Algorithm 1.

In order to express the necessity of feature extraction more clearly, we refine the process of feature extraction, as shown in Figure 5b–k. Figure 5b shows the CSI amplitude data characteristics obtained by the EC method to find the first clipping point on both sides of point a. Figure 5c is a CSI amplitude data characteristic obtained by the second clipping point on both sides of the point a, and the red marked portion in the figure indicates the feature of Figure 5b included. It can be seen from the comparison of the two figures that the feature samples obtained in Figure 5c are clearer than Figure 5b, and subsequent classification is easier. As shown in Figure 5c–f, with the search of the clipping point on both sides of point a, the variation of the obtained CSI amplitude data characteristics is small and complicated. The features obtained in Figure 5b have shown the characteristics of the combined two links. The search is performed in turn, and Figure 5f shows the feature samples obtained from the 10th clipping point on both sides of the point a, which can be seen to be more complicated. Similarly, Figure 5g–k shows the shear point finding process on both sides of point b. When the tenth cut point on both sides of point b is found, it can be seen that the obtained CSI amplitude data feature is more abnormal. Therefore, in order to extract the CSI amplitude data features more representative, we cut at the first clipping point.

**Algorithm 1:** Feature extraction algorithm.

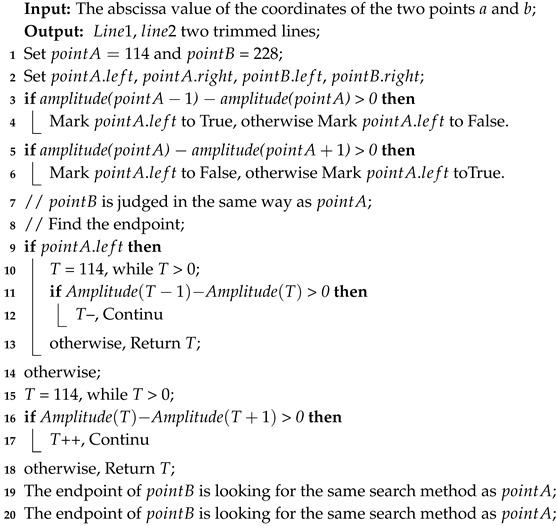



### 3.3. SVM Optimized Positioning Algorithm

The specific implementation of the SVM is as follows. Suppose the sample set has its category represented as $\left(x_{i},y_{i}\right)$i=1,2,⋯,n$y\in \left\{-1,+1\right\}$ are category labels. In the case of linear separability, the support vector classifier attempts to find an optimal classification hyperplane that maximizes the interval $\omega^{T}\cdot x+b=0$.

To find this hyperplane, you need to solve the quadratic programming problem below.
(7)$$\min\frac{1}{2}{∥\omega ∥}^{2}$$
(8)$$y_{i}\left[\left(\omega^{T}\cdot \;x_{i}+b\right)-1\right]\geq 0,i=1,2,\cdots ,n$$

The solution to the above problem can be obtained by solving the following quadratic programming problem.
(9)$$\max W\left(\alpha \right)=\sum_{i=1}^{n}\alpha_{i}-\frac{1}{2}\sum_{i,j=1}^{n}\alpha_{i}\alpha_{j}y_{i}y_{j}x_{i}^{T}x_{j}$$
(10)$$\sum_{i=1}^{l}\alpha_{i}y_{i},0\leq \alpha_{i},i=1,2,\cdots ,n$$

This is a quadratic function optimization problem with inequality constraints, and there is a unique solution. It is easy to prove that only a small part of the Lagrangian multiplier is not zero, and the corresponding sample is the support vector. The resulting optimal classification hyperplane is $f\left(x\right)=\sum_{i=1}^{n}\alpha_{i}^{*}y_{i}x_{i}^{T}x+b^{*}$, where b can be obtained from any of the support vectors. After the optimal hyperplane is obtained, the data classification is completed.

After CSI sample noise reduction and feature extraction, the features need to be classified to complete the positioning. A CSI data sample of 25 positions obtained by the clipping method is shown in Figure 6. It can be seen from the figure that the sample data at different locations is different, and it can be seen that there is similarity in the data samples of each row. From the third behavior limit, it can be seen that the characteristics of the sample data in the first two rows of the upper part are similar and the sample data characteristics in the latter two rows are similar. These similar features, based on CSI sample data, can be classified using the SVM algorithm.

The main process of classifying samples by SVM is that, according to the characteristics of CSI data samples, we can divide the sample positions 1 to 13 into one category, divide 14 to 25 into one category, and then, according to the data collected by the real-time data collected by the positioning position, the sample is judged to be located in the upper part or the lower part. If the upper part is classified in the upper part, you can divide 1 to 7 into one category, divide 8 to 13 into one category, and then judge the position where the data of the specified position is compared with it. The iterative classification is performed in turn, and finally all the CSI sample data can be separated to obtain the position.

## 4. Experimental Study

### 4.1. Experimental Scene

The positioning performance of the algorithm is verified in a real test scenario. As shown in Figure 7b, the test area is a rectangular area having a length and a width of 4.8 m and 5.5 m, respectively, and the rest of the test is divided into 25 small squares of the same size. Two computers are placed on both sides of the test area, one as a receiver and one as a transmitter, and the distance between the two devices placed in the test area is 0.5 m. In the experiment, a computer is required for data processing, to collect experimental data. In order to understand the deployment of the experimental scene more clearly, we made the logic diagram of the experimental scene as shown in Figure 7a.

### 4.2. Experiment Analysis

#### 4.2.1. Impact of the Rate of Packages

In the experimental verification, it is found that the difference in the rate of packet delivery affects its positioning accuracy. The experimental results are shown in Figure 8. When the packet rate is 600 packets per second, the result is optimal and the fluctuation range of its positioning accuracy fluctuates between 1 and 2 m. When the packet transmission rate is 300 packets per second, the positioning accuracy is found to be unstable, and the fluctuation range is between approximately 1.05 m and 1.45 m. This indicates that the CSI data integrity and robustness collected at the packet transmission rate is lower. When the packet delivery rate is 900 packets per second, the stability of the positioning accuracy is also poor. The positioning error reaches ~1.5 m in ~40 s. When the packet transmission rate is 1200 packets per second, the positioning accuracy is relatively stable, but the overall positioning error is significantly increased. This indicates that the data collected at the packet rate is stable, but the noise is increased. Through experimental verification, it can be seen that in the next experimental verification, the accuracy of the experiment can be guaranteed when the delivery rate is maintained at 600 packets per second.

#### 4.2.2. Impact of Single Link and Combined Link

In the EC noise reduction method, the influence of the three links fusion and the single link on the positioning accuracy is different, and the experimental verification result is shown in Figure 9. As can be seen from the figure, the positioning accuracy of the three link fusions is significantly better than the positioning accuracy of a single link, and the trend of the experimental curves of the positioning errors of the three single links (1, 2, and 3) is similar. All four curves experience a slow upward trend with a positioning error of approximately 0 to 1m, after which the four curves have a faster trend until the CDF approaches 1. The experimental results of the three link fusions show that the positioning accuracy is ~85% when the positioning error is ~1.2 m, and the positioning accuracy is ~100% when the positioning error is ~1.95 m. However, the results of three single links show that the positioning accuracy is only 63%, 52%, and 47% when the positioning error is 1.2 m, and the positioning accuracy is only 100% when the positioning error is 2 m.

#### 4.2.3. Impact of the Clipping Facter

In the method description, we mentioned the influence of the clipping factor on the positioning accuracy, so the optimal position of the clipping point is determined by experimental verification.The two independent variables involved in the experiment are the clipping factor and the rate of packet delivery. The experimental verification results are shown in Figure 10. The positioning error is significantly higher at different packet rates and without shear points. When the clipping factor is 1, the positioning error is significantly reduced and the positioning accuracy is optimal at different packet rates.However, as can be seen from the figure, when the packet transmission rate is 300 packets per second, the positioning error fluctuates with the increase of the clipping point, which indicates that the CSI data collected is unstable when the packet transmission rate of is 300 packets per second. At the other three packet rates, after the shear point 1, the positioning error gradually increases with the increase of the shear point and tends to have no positioning error in the case of the shear point.

#### 4.2.4. Comparison of between EC-SVM and SVM

The SVM algorithm involved in the EC-SVM positioning algorithm and the SVM algorithm in the experiment are based on the use of multiclassifiers. However, in this paper, we use the SVM algorithm to classify it under the proposed EC method. Moreover, the classification criteria we use are different, so the positioning accuracy caused by the two algorithms is different. Our classification criteria are that the sample data of 25 positions is first divided into the upper part and the lower part according to the sample characteristics, and then the upper part. Perform a two-division iteration to separate all the data in the upper part. The conventional SVM separates the sample data of 25 positions by, first, separating the position 1 from the others and, sequentially, separating the categories of each position. A comparison experiment between the two methods on the positioning accuracy is shown in Figure 11. As can be seen from the figure, when the number of samples varies from 200 to 1000, the positioning accuracy of the optimized SVM classification is significantly higher than that of the conventional SVM. The positioning accuracy of the optimized SVM is kept between 1.2 m and 1.8 m. The positioning effect is optimal when the number of samples is 600 sets of samples, indicating that the number of samples is 600. The positioning accuracy of the traditional SVM varies from 1.7 m to 2.3 m, and the positioning accuracy is significantly higher.

#### 4.2.5. Overall Performance of EC-SVM Positioning Algorithm

In order to more accurately describe the comparison between EC-SVM and DeepFi, FIFS and Horus, the average error, standard error, and positioning accuracy are compared. The results are shown in Table 1. It can be seen from the figure that the three aspects of the EC-SVM algorithm are superior to the other three algorithms. The standard error of the EC-SVM algorithm is 1.13 m, which indicates that the positioning accuracy of the algorithm is relatively stable. When the positioning error is 1.5 m, the positioning accuracy reaches 85%.

In order to verify the overall positioning performance of the EC-SVM algorithm, this part is compared with the positioning performance of SVR-CSI [32], DeepFi, FIFS and Horus algorithms. The SVR-CSI algorithm is verified in the experimental scenario we set up, and the experimental results obtained are shown in Figure 12. As can be seen from the figure, the positioning accuracy of the EC-SVM positioning algorithm is significantly better than that of SVR-CSI, DeepFi, FIFS, and Horus. The CDF growth trend of the EC-SVM algorithm is relatively fast, and its value soars to ~85% when the positioning error is ~1.2 m, and the CDF of the algorithm reaches 1 at ~1.85 m.The performance of the SVR-CSI algorithm is better than that of the DeepFi, FIFS, and Horus algorithms, but the positioning accuracy is relatively low compared with the EC-SVM algorithm. When the positioning error is ~2.25 m, the CDF value can reach 100%. The performance of the Horus algorithm is relatively the worst, and the growth rate is relatively slow. The CDF reaches ~50% when the positioning error is 1.5 m, and the positioning error reaches 100% when it is ~1.7 m. The positioning performance of the DeepFi and FIFS algorithms is similar, and the trend of the experimental results is approximately similar. The DeepFi algorithm is used when the positioning error is ~100% for a CDF of ~2.65 m, and the FIFS algorithm achieves a positioning accuracy of 100% for a positioning error of ~2.25 m.

## 5. Conclusions

In this paper, a fingerprint indoor positioning algorithm based on EC-SVM is proposed. First, a link fusion and clipping noise reduction method is proposed to establish a highly robust fingerprint database. Then the SVM algorithm is used to classify the data to complete the final positioning. In the experimental verification phase, the performance of the algorithm is verified by five parts: the impact of the rate of packages on positioning accuracy, the influence of three link fusion and single link on positioning accuracy, the impact of the rate of packages and clipping fact on positioning accuracy, EC-SVM and traditional SVM comparison and EC-SVM algorithm, and SVR-CSI, DeepFi, FIFS, and Horus algorithm comparison experiment. The experimental results show that the positioning accuracy of the EC-SVM algorithm is 85% when the positioning error is 1.2 m, and its positioning accuracy is better than that of SVR-CSI, DeepFi, FIFS, and Horus.

## Figures and Tables

**Figure 1 sensors-19-03689-f001:**
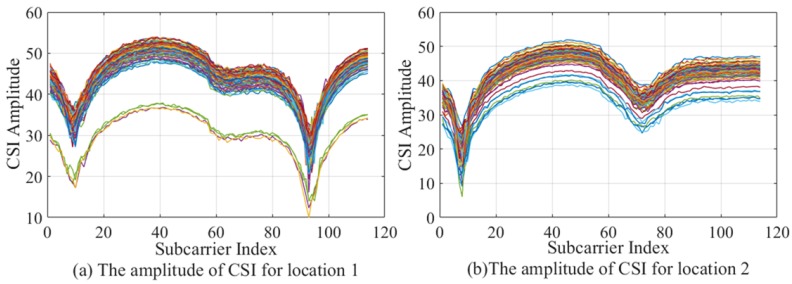
Channel State Information (CSI) amplitude at different locations.

**Figure 2 sensors-19-03689-f002:**
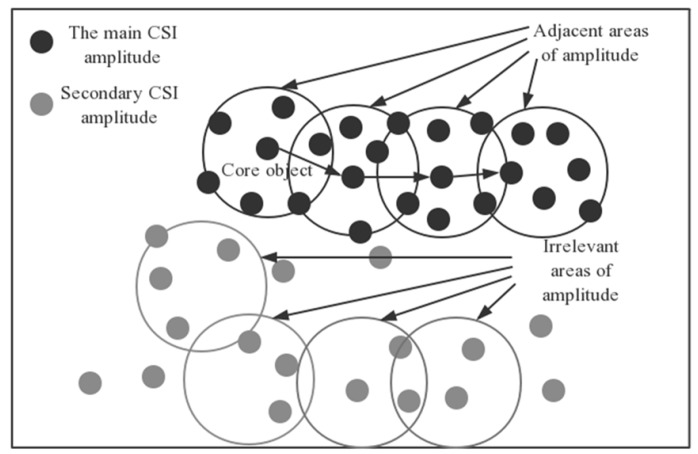
Density-Based Spatial Clustering of Applications with Noise (DBSCAN) algorithm principle.

**Figure 3 sensors-19-03689-f003:**
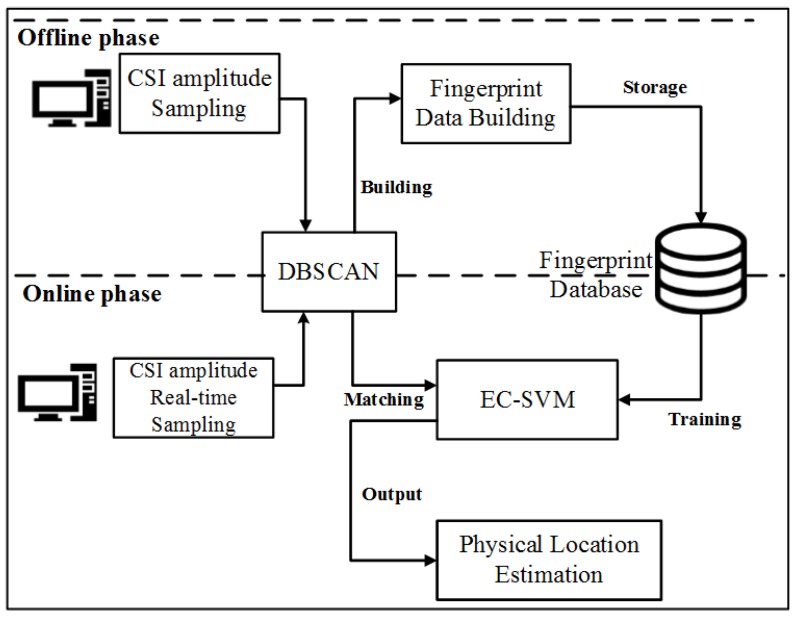
Localization flowchart.

**Figure 4 sensors-19-03689-f004:**
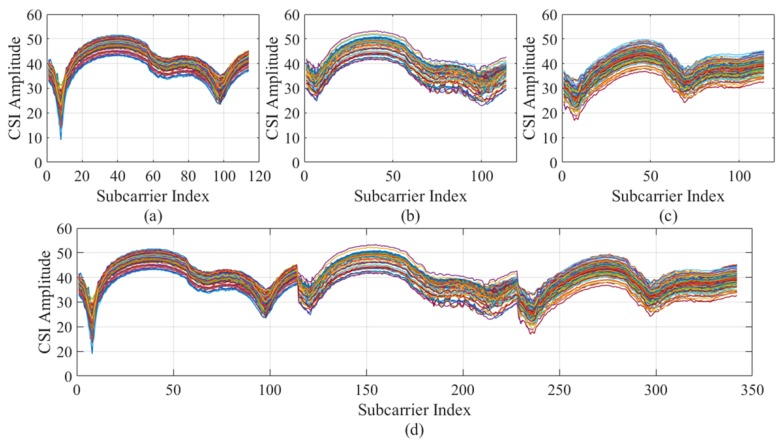
Link fusion process: (**a**–**c**) single link and (**d**) combined link.

**Figure 5 sensors-19-03689-f005:**
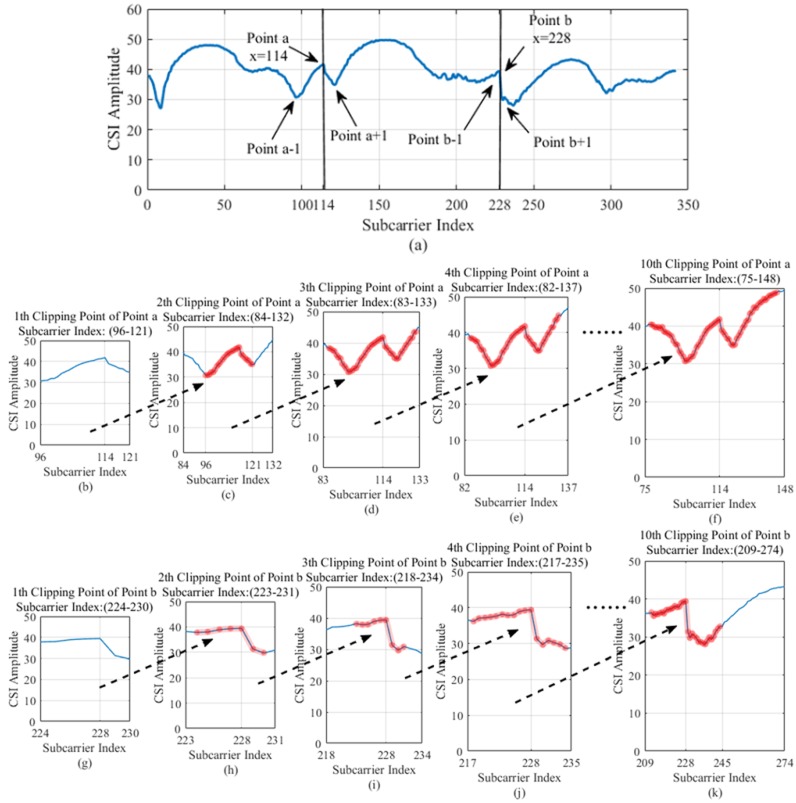
Characteristics of CSI amplitude extract. (**a**) Representation of the two clipping base points of a and b of the fusion link. (**b**,**g**) CSI amplitude characteristics obtained from the first clipping point. (**c**,**h**) CSI amplitude characteristics obtained from the second clipping point. (**d**,**i**) CSI amplitude characteristics obtained from the third clipping point. (**e**,**j**) CSI amplitude characteristics obtained from the forth clipping point. (**f**,**k**) CSI amplitude characteristics obtained from the tenth clipping point.

**Figure 6 sensors-19-03689-f006:**
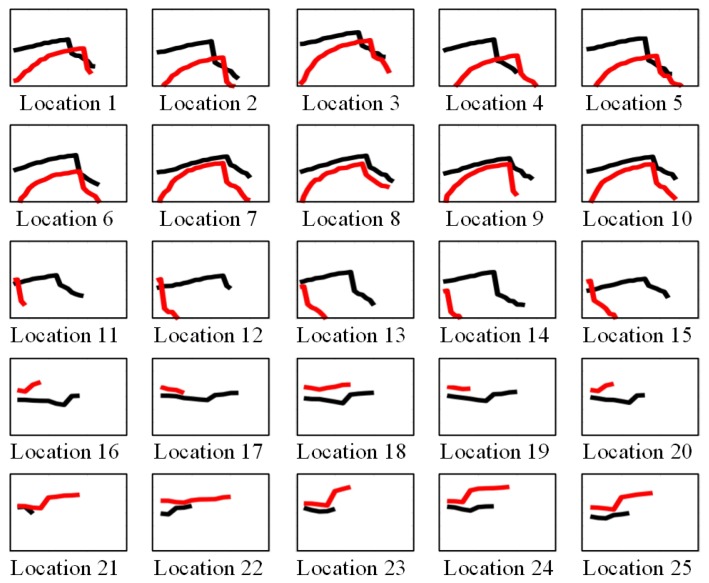
Extracted CSI amplitude sample.

**Figure 7 sensors-19-03689-f007:**
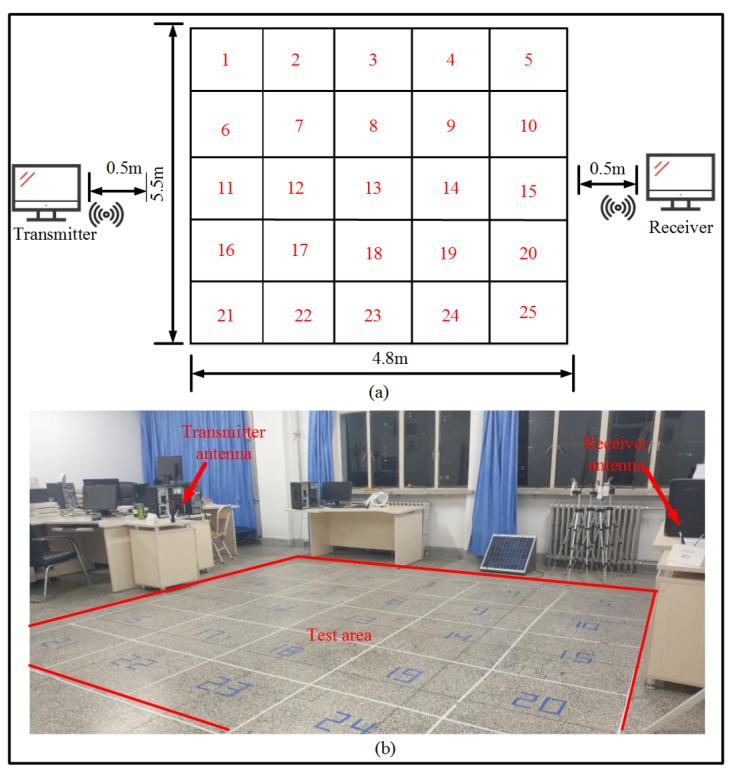
Experimental scene: (**a**) Experimental scene logic diagram (**b**). Real experimental site deployment.

**Figure 8 sensors-19-03689-f008:**
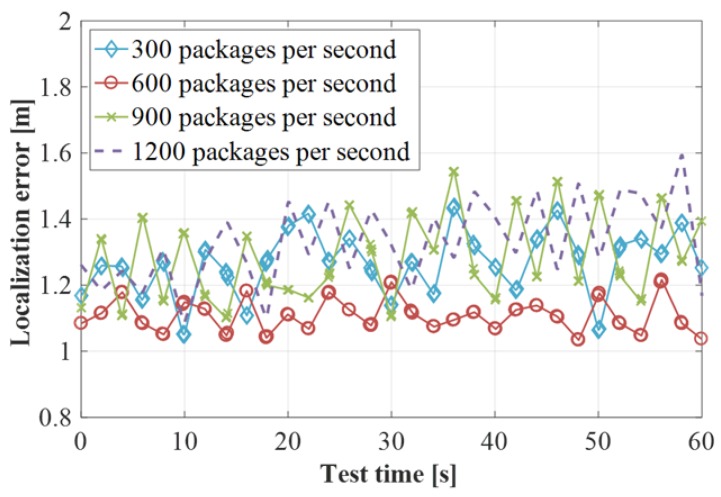
Change the impact of the packet rate on the positioning error.

**Figure 9 sensors-19-03689-f009:**
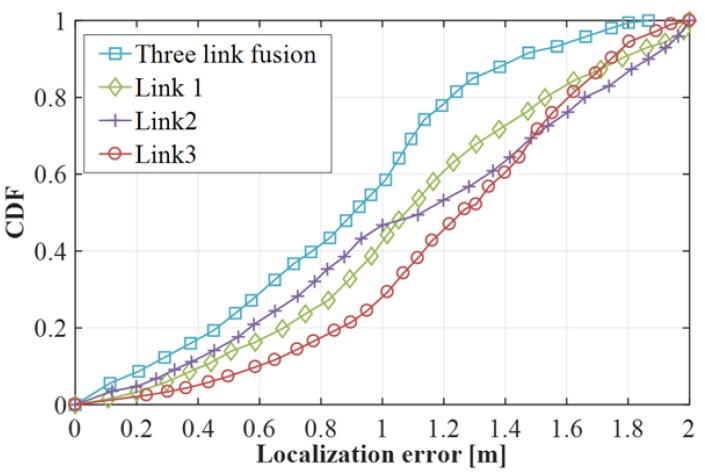
Performance comparison between single link and combined link.

**Figure 10 sensors-19-03689-f010:**
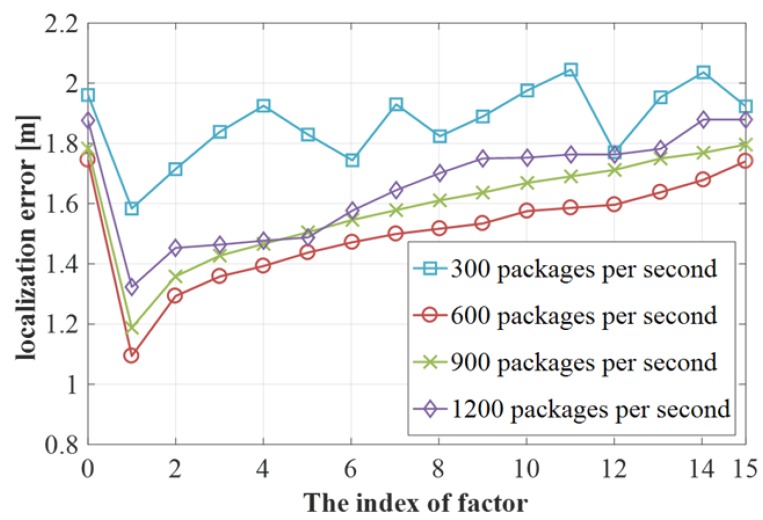
Impact of the clipping factor.

**Figure 11 sensors-19-03689-f011:**
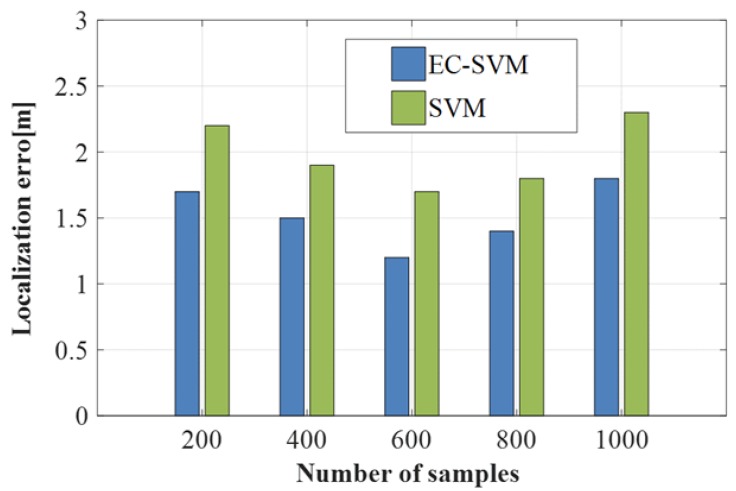
Comparison of optimized SVM with traditional SVM.

**Figure 12 sensors-19-03689-f012:**
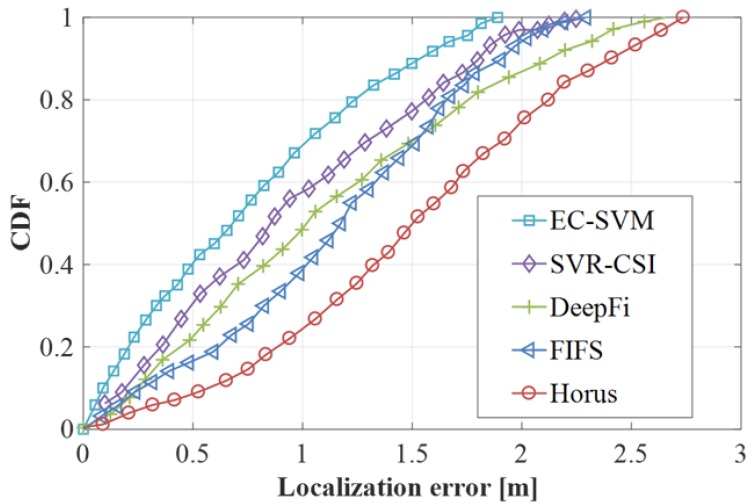
Comparison of positioning accuracy of four algorithms.

**Table 1 sensors-19-03689-t001:** Comparison of algorithm positioning performance.

Algorithms	Average Error	Standard Error	Positioning Accuracy (1.5 m)
EC-SVM	1.37	1.13	89.00%
DeepFi	1.53	1.27	69.41%
FIFS	1.86	1.42	68.74%
Horus	2.61	1.77	50.48%

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
