# Peer review of "Endpoints-Clipping CSI Amplitude for SVM-Based Indoor Localization"

_sensors, 2019, doi:10.3390/s19173689_

Round 1

Reviewer 1 Report

This paper presents a CSI based indoor localization technique with noise reduction enhancement. My comments are listed as follows.

1, The background search is insuffcient. It does not include some latest works on this area. For example,

Wu Z, Xu Q, Li J, et al. Passive Indoor Localization Based on CSI and Naive Bayes Classification[J]. IEEE Transactions on Systems,Man,and Cybernetics: Systems, vol.48, no.9, pp.1566-1577,2018.

Wu, Zhefu,Jiang, Lei,Jiang, Zhuangzhuang,et al.Accurate Indoor Localization Based on CSI and Visibility Graph[J].SENSORS,vol.18,no.9,pp.2549-2567,2018.

2, The contribution is unclear. The authors should statr clearly in the introducion about the main contribution and novelty of the paper.

3, The authors should put more explanation about their methods. For example, in page 6, line 143, the authors claim the result is optimal. Why it is optimal? It is optimal against what? Is there any proof about this optimality? Moreover, for the search of the clipping point, the author should also give more explanation about why finding such a clipping point can help to reduce the CSI noises.

4, The experiment results need to be improved. The authors should compare their results with state-of-art indoor CSI based localization technique.

Author Response

Response to Reviewer 1 Comments

The authors would like to thank reviewers and editor for their careful reviews and constructive criticism on our manuscript sensors-546333, which have helped us revise and improve the quality of the manuscript. Based on these helpful comments, we have revised this paper carefully. At the same time, we try our best to enhance the presentation, and some flaws are corrected. All of the corrections made in the revision are marked in red. We hope that it helps editor and reviewers finding the corrections. In the following, we address all the comments and suggestions of reviewers one-by-one.

Comments and Suggestions for Authors

This paper presents a CSI based indoor localization technique with noise reduction enhancement. My comments are listed as follows.

Point 1: The background search is insuffcient. It does not include some latest works on this area.  For example,

Wu Z, Xu Q, Li J, et al. Passive Indoor Localization Based on CSI and Naive Bayes Classification[J]. IEEE Transactions on Systems,Man,and Cybernetics: Systems, vol.48, no.9, pp.1566-1577,2018.

Wu, Zhefu,Jiang, Lei,Jiang, Zhuangzhuang,et al.Accurate Indoor Localization Based on CSI and Visibility Graph[J].SENSORS,vol.18,no.9,pp.2549-2567,2018.

Response 1: Thank the reviewer for this constructive suggestion. Your comments are extremely valuable, in the background description of the introduction, we did miss some of the latest related work. Therefore, in order to ensure the fullness of the manuscript background description, we added and revised the introduction, and added the following references on the basis of supplements.

Page 2, line 15: We have added content “A passive indoor positioning technology based on 49 CSI and naive Bayes is proposed in [15], which improves the naive Bayesian algorithm to make the positioning result more accurate. The experimental part verifies the performance of the positioning algorithm in a variety of environments. Wu,Zhefu, Jiang,et al. [16]proposed a passive indoor positioning  algorithm combined with mathematical statistics and networks, mainly by improving the visibility graph (VG) technology to establish a complex indoor network to achieve better positioning accuracy.”

We have added references [15] and [16] as follows:

[15] Wu Z, Xu Q, Li J, et al. Passive Indoor Localization Based on CSI and Naive Bayes Classification[J]. IEEE Transactions on Systems,Man,and Cybernetics: Systems, vol.48, no.9, pp.1566-1577,2018.

[16] Wu, Zhefu,Jiang, Lei,Jiang, Zhuangzhuang,et al.Accurate Indoor Localization Based on CSI and Visibility Graph[J].SENSORS,vol.18,no.9,pp.2549-2567,2018.

Point 2: The contribution is unclear. The authors should statr clearly in the introduction about the main contribution and novelty of the paper.

Response 2: Thank the reviewer for this constructive suggestion. The modifications made to enhance the logic of the introduction and the contributions of this article are as follows:

The last sentence of the second paragraph of the introduction “Since the CSI signal is sensitive, it is necessary to propose a better noise reduction method.” has updated to “In summary, ensuring the robustness while improving the positioning accuracy is an urgent problem to be solved.”

We added contributions to the manuscript in the introduction“In general, in this work, we have made the following main contributions:

We propose a novel indoor fingerprint localization algorithm. According to the characteristics of the communication link, the collected CSI data is denoised by the DBSCAN algorithm in the online and offline phases. We propose an EC noise reduction method, which firstly integrates three CSI communication links, and then performs feature extraction on the fused links to obtain a robust fingerprint database. We validate the proposed theory and method in a real experimental scenario. The experimental results show that the proposed positioning algorithm is superior to its comparison algorithm and has high robustness”

Point 3: The authors should put more explanation about their methods. For example, in page 6, line 143, the authors claim the result is optimal. Why it is optimal? It is optimal against what? Is there any proof about this optimality? Moreover, for the search of the clipping point, the author should also give more explanation about why finding such a clipping point can help to reduce the CSI noises.

Response 3: We sincerely thank the reviewer for the very professional suggestions. Your comments are extremely valuable, making the content of our manuscript more substantial and more rigorous.  Regarding the EC method, we have not described it clearly and comprehensively. We have supplemented and revised the content of this part. The search result of the cut point is why it is optimal to determine a point. In the experimental part, we also added a set of experiments to prove it.

Page 4, the top of Figure 5: “In order to reduce the noise of the data, we propose an EC method. As shown in Fig. 5, two points a and b are determined according to the joints of the three links, the abscissa of point a is x=56, and the abscissa of point b is x=112. The point on the left side of point a is recorded as a-1and the point on the right is recorded as a+1; the point b on the left side of point b is recorded as b-1 and the point on the right side is recorded as b+1. The algorithm mainly searches for the shear point, determines whether the trend of the points on the left and right sides of the two points a and b is monotonous. By verifying, the first extreme value at both ends of points a and b can be found as the clipping point, and the result is optimal, so the point is finally used as the clipping point. The pseudo code of the specific clipping process is given in Algorithm 1.” are replaced by “We have a simple combination of the three data links, although the data link we use the DBSCAN algorithm for noise reduction, but there is still noise in the data. In order to further optimize the CSI data, we propose a novel EC noise reduction method. In order to express the steps of the EC algorithm more clearly, we randomly take a CSI amplitude data from the combined CSI data link as shown in Figure5. We use the splicing points a, b of the link as the base point for finding the clipping point. We define the clipping factor sequences for the two points a and b as $\alpha $ and $\beta $ respectively:

\[\begin{align}

  & \alpha =\left\{ \begin{align}

  & \left[ \left( a-1 \right),\ \left( a-2 \right),\left( a-3 \right),\cdots \left( a-{{n}_{1}} \right) \right],\ {{n}_{1}}\in \left( 0,114 \right) \\

 & \left[ \left( a+1 \right),\ \left( a+2 \right),\left( a+3 \right),\cdots \left( a+{{n}_{2}} \right) \right],\ {{n}_{2}}\in (114,228) \\

\end{align} \right. \\

 &  \\

\end{align}\]

\[\begin{align}

  & \beta =\left\{ \begin{align}

  & \left[ \left( b-1 \right),\ \left( b-2 \right),\left( b-3 \right),\cdots \left( b-{{n}_{3}} \right) \right],\ {{n}_{3}}\in \left( 114,228 \right) \\

 & \left[ \left( b+1 \right),\ \left( b+2 \right),\left( b+3 \right),\cdots \left( b+{{n}_{4}} \right) \right],\ {{n}_{4}}\in (228,342) \\

\end{align} \right. \\

 &  \\

\end{align}\]

Where $a-1$ represents the first clipping point found to the left of point a;  $a+1$ represents the first clipping point found to the right of point a;   $b-1$ represents the first shear point found to the left of point b;    $b+1$represents the first shear point found to the right of point b.

First, we look for the clipping points on both sides of point a. Starting from point a to the left, the monotonicity of the curve is judged by the difference of adjacent points until its monotonicity changes. We take the adjacent point where monotonicity changes as the first clipping point to be found. Since the curve trend of the CSI amplitude data is wavy, we will find many eligible shear points. For the same reason, we can also find many cut points on the left side of point a. However, through experimental verification analysis, we obtained that the sample data formed by the first shear point among the many shear points found on both sides of point a is better. Since the curve trend of the CSI data is wavy, we will find many eligible clipping points. For the same reason, we can also find many clipping points on the left side of point a. However, through experimental verification analysis, we obtained that the sample data formed by the first clipping point among the many shear points found on both sides of point a is better. After determining the clipping points on both sides of point a, we take the data between the two clipping points as the first feature sample at a certain position. In the same way, we can find the shear points on both sides of point b to obtain another feature sample at the same position. The samples obtained at two points a and b are sample features at one location. The feature sample data of the 25 positions obtained by the EC method is as shown in Figure 6. The pseudo code of the specific clipping process is given in Algorithm 1.”.

Point 4: The experiment results need to be improved. The authors should compare their results with state-of-art indoor CSI based localization technique.

Response 4: We sincerely thank the reviewer for the very professional suggestions. In the experimental comparison, we did not compare with the state-of-art indoor positioning algorithms, so based on your professional opinion, we added a set of experiments in the overall performance test comparison. The experimental results are shown in Figure 12. And we have adjusted the structure of the experimental part to ensure its logic.

Once again, we thank reviewers and editor very much for your constructive comments and suggestions.

Reviewer 2 Report

This paper proposed the algorithm of Endpoints-Clipping CSI amplitude and Support Vector Machine based on CSI for indoor positioning. The EC-SCM indoor localization focused on using the combination of communicating link for noise reduction and feature extraction and using a binary tree structure multi-classes SVM classifier to obtain the position. However, there are some points which may worth being noticed.

1.     There are logical problems in some sentences, and the structure of this paper is not reasonable.

2.     There should be more detailed description in EC section. It is better to compare with other features to verify the necessity of the selected feature in the manuscript.

3.     The section of SVM cannot be considered as an innovation, which is just a binary tree structure multi-classes SVM classifier.

4.     The experimental results can be analyzed from more aspects to verify the feasibility of the proposed method in the manuscript.

Author Response

Response to Reviewer 2 Comments

The authors would like to thank reviewers and editor for their careful reviews and constructive criticism on our manuscript sensors-546333, which have helped us revise and improve the quality of the manuscript. Based on these helpful comments, we have revised this paper carefully. At the same time, we try our best to enhance the presentation, and some flaws are corrected. All of the corrections made in the revision are marked in red. We hope that it helps editor and reviewers finding the corrections. In the following, we address all the comments and suggestions of reviewers one-by-one.

Comments and Suggestions for Authors

This paper proposed the algorithm of Endpoints-Clipping CSI amplitude and Support Vector Machine based on CSI for indoor positioning. The EC-SCM indoor localization focused on using the combination of communicating link for noise reduction and feature extraction and using a binary tree structure multi-classes SVM classifier to obtain the position. However, there are some points which may worth being noticed.

Point 1:    There are logical problems in some sentences, and the structure of this paper is not reasonable.

Response 1: We sincerely thank the reviewer for the very professional suggestions.

(1) Based on your valuable comments, we have modified the language logic and expression of some sentences.

Page 1, Abstract, line 2: “Because the flexibility of the CSI signals,” is replaced by “Due to the flexibility of the CSI signals”.

Page 1, Abstract, line 4: “For this problem” is updated to “To the tackle”.

Page 1, Abstract, the last line: “The experimental results show that the positioning
 performance of the EC-SVM algorithm is superior.” is replaced by “The experimental results show that the positioning accuracy of the EC-SVM algorithm is superior to state-of-art indoor CSI based localization technique.”

Page 1,0. Introduction, line 2: “wireless indoor positioning technology has been researched in different way”, “way” is corrected to “ways”.

Page 1, 0. Introduction, paragraph 1, last sentences: “In one word, the indoor positioning technology based on CSI has attracted more and more attention in recent years”, “In a word” is updated to “Therefore”.

Page 7, paragraph 3, line 1: “First, we look for the clipping points on both sides of point a.”, “look for” is updated to “search”.

   (2) We have also adjusted the logical structure of the manuscript.

    1) Page 7, 2.2.2 Characteristic of combined communication extract: The content after the third paragraph is adjusted to the beginning of the chapter.

    2)Page 10, 3.2 Experiment Analysis: The order of the experimental parts was adjusted.

The first experiment was updated to determine the impact of the packet rate on the positioning accuracy.

The second experiment was updated to a CDF plot of link comparisons.

The third experiment was updated as the effect of the shear factor on the positioning accuracy.

Thank you very much for your comments and make our manuscript more logical.

Point 2: There should be more detailed description in EC section. It is better to compare with other features to verify the necessity of the selected feature in the manuscript.

Response 2: We sincerely thank the reviewer for the very professional suggestions. Your comments are extremely valuable, making the content of our manuscript more substantial and more rigorous.  Regarding the EC method, we have not described it clearly and comprehensively. We have supplemented and revised the content of this part. The search result of the cut point is why it is optimal to determine a point. In the experimental part, we also added a set of experiments to prove it.

Page 4, the top of Figure 5: “In order to reduce the noise of the data, we propose an EC method. As shown in Fig. 5, two points a and b are determined according to the joints of the three links, the abscissa of point a is x=56, and the abscissa of point b is x=112. The point on the left side of point a is recorded as a-1and the point on the right is recorded as a+1; the point b on the left side of point b is recorded as b-1 and the point on the right side is recorded as b+1. The algorithm mainly searches for the shear point, determines whether the trend of the points on the left and right sides of the two points a and b is monotonous. By verifying, the first extreme value at both ends of points a and b can be found as the clipping point, and the result is optimal, so the point is finally used as the clipping point. The pseudo code of the specific clipping process is given in Algorithm 1.” are replaced by “We have a simple combination of the three data links, although the data link we use the DBSCAN algorithm for noise reduction, but there is still noise in the data. In order to further optimize the CSI data, we propose a novel EC noise reduction method. In order to express the steps of the EC algorithm more clearly, we randomly take a CSI amplitude data from the combined CSI data link as shown in Figure5. We use the splicing points a, b of the link as the base point for finding the clipping point. We define the clipping factor sequences for the two points a and b as $\alpha $  and $\beta $ respectively:

\[\begin{align}

  & \alpha =\left\{ \begin{align}

  & \left[ \left( a-1 \right),\ \left( a-2 \right),\left( a-3 \right),\cdots \left( a-{{n}_{1}} \right) \right],\ {{n}_{1}}\in \left( 0,114 \right) \\

 & \left[ \left( a+1 \right),\ \left( a+2 \right),\left( a+3 \right),\cdots \left( a+{{n}_{2}} \right) \right],\ {{n}_{2}}\in (114,228) \\

\end{align} \right. \\

 &  \\

\end{align}\]

\[\begin{align}

  & \beta =\left\{ \begin{align}

  & \left[ \left( b-1 \right),\ \left( b-2 \right),\left( b-3 \right),\cdots \left( b-{{n}_{3}} \right) \right],\ {{n}_{3}}\in \left( 114,228 \right) \\

 & \left[ \left( b+1 \right),\ \left( b+2 \right),\left( b+3 \right),\cdots \left( b+{{n}_{4}} \right) \right],\ {{n}_{4}}\in (228,342) \\

\end{align} \right. \\

 &  \\

    Where $a-1$ represents the first clipping point found to the left of point a;  $a+1$ represents the first clipping point found to the right of point a;   $b-1$ represents the first shear point found to the left of point b;    $b+1$represents the first shear point found to the right of point b.

    First, we look for the clipping points on both sides of point a. Starting from point a to the left, the monotonicity of the curve is judged by the difference of adjacent points until its monotonicity changes. We take the adjacent point where monotonicity changes as the first clipping point to be found. Since the curve trend of the CSI amplitude data is wavy, we will find many eligible shear points. For the same reason, we can also find many cut points on the left side of point a. However, through experimental verification analysis, we obtained that the sample data formed by the first shear point among the many shear points found on both sides of point a is better. Since the curve trend of the CSI data is wavy, we will find many eligible clipping points. For the same reason, we can also find many clipping points on the left side of point a. However, through experimental verification analysis, we obtained that the sample data formed by the first clipping point among the many shear points found on both sides of point a is better. After determining the clipping points on both sides of point a, we take the data between the two clipping points as the first feature sample at a certain position. In the same way, we can find the shear points on both sides of point b to obtain another feature sample at the same position. The samples obtained at two points a and b are sample features at one location. The feature sample data of the 25 positions obtained by the EC method is as shown in Figure 6. The pseudo code of the specific clipping process is given in Algorithm 1.”.

Point 3: The section of SVM cannot be considered as an innovation, which is just a binary tree structure multi-classes SVM classifier.

Response 3: We sincerely thank the reviewer for the very professional suggestions. You are great right that SVM multi-classifier is not a point of innovation, but the classification criteria of multi-classifiers can affect the performance of the classification to a certain extent. By extracting the features of the combined link, we derive the features at 25 locations as shown in Figure 6. Through the change of CSI data, we manually divide the site into the first search area, the second search area, etc. according to the overall change trend of the CSI data in the site. Finally, the SVM multi-classification function was used to classify the site.

Point 4: The experimental results can be analyzed from more aspects to verify the feasibility of the proposed method in the manuscript.

Response 4: We sincerely thank the reviewer for the very professional suggestions.You are great correct that we really need to add content in the experimental part, to ensure the enrichment of the article, so we added a set of experiments on the impact of the shear factor on the positioning accuracy, the experimental results are shown in Figure 10. And the structure of the experimental part was adjusted to ensure its logic.

Once again, we thank reviewers and editor very much for your constructive comments and suggestions.

Reviewer 3 Report

The paper is interesting and present a promising technique, however some issues should be clarified and the presentation style should be inproved for the sake of readablity. 

- Please check whether all acronyms are properly defined on their first usage (RSS, GPS, MAC, EC-SVM...)

- Page 3, two lines above (3), please verify the value and the measurement units 40 mHZ. Was it meant to be 40 MHz? 

- The subcarrier indexes in Fig. 4 are a bit confusing. In Figs 4a,b, and c the subcarrier indexes [0:60] in the 3 figures seemingly refer to same frequency band. In Fig. 4d the same responses are seemingly allocated in 3 different bands. This should be clarified, especially for readers unfamiliar with the DBSCAN algorithm.

- Fig. 8-9: the metric localization error/m is not very clear. Did you mean that the localization error is expressed in meters? If so please consider changing it into "localization error [m]" for the sake of readability. If not so please introduce the metric in the text.

- Fig. 9: why is the localization error increasing with the number of tests? 

- Fig. 10: is OP-SVM a synonimous of EC-SVM? If so, please consider using a single name throughout the paper.  

Author Response

Response to Reviewer 3 Comments

The authors would like to thank reviewers and editor for their careful reviews and constructive criticism on our manuscript sensors-546333, which have helped us revise and improve the quality of the manuscript. Based on these helpful comments, we have revised this paper carefully. At the same time, we try our best to enhance the presentation, and some flaws are corrected. All of the corrections made in the revision are marked in red. We hope that it helps editor and reviewers finding the corrections. In the following, we address all the comments and suggestions of reviewers one-by-one.

Comments and Suggestions for Authors

The paper is interesting and present a promising technique, however some issues should be clarified and the presentation style should be improved for the sake of readability.

Response: We sincerely thank the reviewer for the very professional suggestions. There is indeed a problem with the expression style in our manuscript, so based on your extremely valuable, we have checked and corrected the expression style of the full text.

Point 1: Please check whether all acronyms are properly defined on their first usage (RSS, GPS, MAC, EC-SVM...)

Response 1: We sincerely thank the reviewer for the very professional suggestions. Your comments are extremely valuable, we have checked and corrected all the acronyms, the details of the specific changes are as follows:

Page 1, the first paragraph of the introduction: Line 3 “At present, indoor positioning technologies appearing at home and abroad generally include GPS”, “GPS” has updated to “Global Positioning System (GPS)”.

 Line12 “However, due to the instability and inactivity of RSS signal”, “RSS” has updated to “Received Signal Strength (RSS)”.

Page 2, line 4 “RSSI can be obtained from ordinary Wi-Fi receiving devices, which directly reflects the power strength from the transceiver”, “RSSI” has updated to “RSS signal”.

Line 6 “However, for CSI, the RSSI only indicates the total received energy of the channel,”,“RSSI” has updated to “RSS”

Page 3, 1. System Model and Relevant Definitions. Paragraph 1: Line 1-3 “The traditional WLAN-based passive indoor positioning method is mainly based on RSS. Due to the shortcomings of RSS signal, such as poor signal stability and poor reliability, the actual positioning effect is not ideal. Compared to RSS, CSI provides more granular channel information,”,“RSSI” has updated to “RSS”. Page 3, 1. System Model and Relevant Definitions. Paragraph 2. “OFDM” has updated to“Orthogonal Frequency Division Multiplexing (OFDM)”. Page 3, 1. System Model and Relevant Definitions. Paragraph 4. “Orthogonal Frequency Division Multiplexing (OFDM)” has updated to “OFDM”. “MIMO” has updated to “Multiple-Input Multiple-Output (MIMO)”. Page 5, Paragraph 1, line 1 “Support Vector Machines (SVM)” has updated to “SVM”.

         Line 2 “support vector machines” has updated to“SVM”.

Point 2: Page 3, two lines above (3), please verify the value and the measurement units 40 mHZ. Was it meant to be 40 MHz?

Response 2: We sincerely thank the reviewer for the very professional suggestions. I am very sorry that due to our negligence, the writing format of 40mHZ in our manuscript is wrong, we have modified it to the correct format.

Page 3 ,1. System Model and Relevant Definitions, Paragraph 4. “40mHZ” has corrected to “40MHz”. Since the number of carriers depends on the carrier bandwidth, the specification indicates that the number of carriers is 114 at 40MHz. As shown in the figure below, the number of carriers in the first group of 40 MHz is 114. Therefore, the Figure 4 has been corrected.

Point 3: The subcarrier indexes in Fig. 4 are a bit confusing. In Figs 4a,b, and c the subcarrier indexes [0:60] in the 3 figures seemingly refer to same frequency band. In Fig. 4d the same responses are seemingly allocated in 3 different bands. This should be clarified, especially for readers unfamiliar with the DBSCAN algorithm.

Response 3: We sincerely thank the reviewer for the very professional suggestions.

Regarding the frequency bandwidth problem in Figure 4, the four pictures are all in the same frequency bandwidth. Since we simply flatten together Figures 4(a), (b) and (c), the resulting Figure 4(d) is also the frequency bandwidth in which (a), (b) and (c) are located. The reason for the graphics fusion is for the processing of subsequent algorithms. The description of the DBSCAN algorithm in the manuscript is not clear enough, and we have supplemented and modified it according to the reviewer's extremely valuable

Page 4, the paragraph below Figure 1: “CSI amplitude data will have abnormal values due to multipath and other effects. In order to ensure accuracy, the DBSCAN algorithm is used to eliminate the abnormal parts.” is replaced by “It can be seen from the Figure 1(a),(b) that most of the CSI amplitude values are in a cluster of closely adjacent states, but some of the data is obviously far away from the data cluster set, and the part of the data is called abnormal value. To ensure accuracy, the DBSCAN algorithm can be used to eliminate the abnormal parts.”

Page 4, the paragraph below Figure 1: We have added some content at the end of the paragraph “It can be seen from Figure 2 that the CSI amplitude data distributed in the upper part shows a high degree of correlation, indicating that the data of this part belong to one category. The CSI amplitude data in the lower part is loosely dispersed, and its full correlation cannot be guaranteed. If the part of the data cannot be attributed to the same category, it is called an outlier. Therefore, the outliers can be eliminated by the judgment of the method to ensure the accuracy of the data.”

Point 4: Fig. 8-9: the metric localization error/m is not very clear. Did you mean that the localization error is expressed in meters? If so please consider changing it into "localization error [m]" for the sake of readability. If not so please introduce the metric in the text.

Response 4: We sincerely thank the reviewer for the very professional suggestions. We have corrected all the metric localization error/m in figure.

Fig. 8-11: all “localization error/m” have corrected to “localization error [m]”

Point 5: Fig. 9: why is the localization error increasing with the number of tests?

Response 5: We sincerely thank the reviewer for the very professional suggestions. I am very sorry, due to our negligence and carelessness, the manuscript has been incorrectly displayed in the layout process. According to your professional comments, we have uploaded the correct version of the figure. The correct version is shown in Figure 8. And in order to ensure the logic of the experimental part of the manuscript, we have adjusted the structure of the experimental part.

Point 6: Fig. 10: is OP-SVM a synonimous of EC-SVM? If so, please consider using a single name throughout the paper.  

Response 6: We sincerely thank the reviewer for the very professional suggestions. Since our mistakes do not have a uniform naming approach, we have modified the legend in Figure 10.

Fig.10: “OP-SVM” has updated to “EC-SVM”.

Once again, we thank reviewers and editor very much for your constructive comments and suggestions.

Round 2

Reviewer 2 Report

The authors have all been appropriately revised based on the previous suggestions. 

The sentences of the article becomes logical and the structure of the article is more reasonable relatively.

Authors made detailed explanation on EC section. But it's still not sufficient enough to verify the necessity of the selected feature.

Authors also made some appropriate adjustments about illustration of  innovation about using the SVM approach.

In experiment section, authors added a set of experiments the impact of the shear factor on the positioning accuracy to verify the feasibility of the proposed method in the manuscript.

Author Response

Response to Reviewer 2 Comments

The authors would like to thank reviewers and editor for their careful reviews and constructive criticism on our manuscript sensors-546333, which have helped us revise and improve the quality of the manuscript. Based on these helpful comments, we have revised this paper carefully. At the same time, we try our best to enhance the presentation, and some flaws are corrected. All of the corrections made in the revision are marked in red. We hope that it helps editor and reviewers finding the corrections. In the following, we address all the comments and suggestions of reviewers one-by-one.

Comments and Suggestions for Authors

The authors have all been appropriately revised based on the previous suggestions.

The sentences of the article becomes logical and the structure of the article is more reasonable relatively.

Authors made detailed explanation on EC section. But it's still not sufficient enough to verify the necessity of the selected feature.

Authors also made some appropriate adjustments about illustration of innovation about using the SVM approach.

In experiment section, authors added a set of experiments the impact of the shear factor on the positioning accuracy to verify the feasibility of the proposed method in the manuscript.

Point 1: Authors made detailed explanation on EC section. But it's still not sufficient enough to verify the necessity of the selected feature.

Response: Thank you very much for your affirmation of our previous work and for the very professional comments. We are not sufficiently good enough to describe the EC method, and we have already modified and added it.

Page 8, Figure 5: We added a set of diagrams to explain the process of feature extraction.

Page 8, Figure 5 below: We have added a description of the figure: “ In order to express the necessity of feature extraction more clearly, we refine the process of feature extraction, as shown in Figure5(b)-(k).Figure5(b) shows the CSI amplitude data characteristics obtained by the EC method to find the first clipping point on both sides of point a.Figure5(c) is a CSI amplitude data characteristic obtained by the second clipping point on both sides of the point a, and the red marked portion in the figure indicates the feature of Figure 5(b) included. It can be seen from the comparison of the two figures that the feature samples obtained in Figure5(c) are clearer than figure5(b), and subsequent classification is easier. As shown in Figure5(c)-(f), with the search of the clipping point on both sides of point a, the variation of the obtained CSI amplitude data characteristics is small and complicated. The features obtained in Figure5(b) have shown the characteristics of the combined two links. The search is performed in turn, and Figure5(f) shows the feature samples obtained from the 10th clipping point on both sides of the point a, which can be seen to be more complicated. Similarly, Figures 5(g)-(k) show the shear point finding process on both sides of point b. When the tenth cut point on both sides of point b is found, it can be seen that the obtained CSI amplitude data feature is more abnormal. Therefore, in order to extract the CSI amplitude data features more representative, we cut at the first clipping point.”

Once again, we thank reviewers and editor very much for your constructive comments and suggestions.

Reviewer 3 Report

Authors carefully revised the paper, adequately answering all critical remarks.
Please consider that a "40mHz" survived between (2) and (3), and should probably be corrected into "40 MHz".

Author Response

Response to Reviewer 3 Comments

The authors would like to thank reviewers and editor for their careful reviews and constructive criticism on our manuscript sensors-546333, which have helped us revise and improve the quality of the manuscript. Based on these helpful comments, we have revised this paper carefully. At the same time, we try our best to enhance the presentation, and some flaws are corrected. All of the corrections made in the revision are marked in red. We hope that it helps editor and reviewers finding the corrections. In the following, we address all the comments and suggestions of reviewers one-by-one.

Comments and Suggestions for Authors

Authors carefully revised the paper, adequately answering all critical remarks.

Please consider that a "40mHz" survived between (2) and (3), and should probably be corrected into "40 MHz".

Response: Thank you very much for your affirmation of our previous work and for the very professional comments.

 Page 3 ,1. System Model and Relevant Definitions, Paragraph 4. “40mHZ” has corrected to “40MHz”.

Once again, we thank reviewers and editor very much for your constructive comments and suggestions.